# ACTION SEQUENCE PLANNER: AN ALTERNATIVE FOR OFFLINE REINFORCEMENT LEARNING

## ABSTRACT

Offline reinforcement learning methods, which typically train agents that make decisions step by step, are known to suffer from instability due to bootstrapping and function approximation, especially when applied to tasks requiring long-horizon planning. To alleviate these issues, in this paper, we propose a novel policy gradient approach by planning an action sequence in a high-dimensional space. This design implicitly models temporal dependencies, excelling in long-horizon and horizon-critical tasks. Furthermore, we discover that replacing maximum likelihood with cross-entropy loss in policy gradient methods significantly stabilizes training gradients, leading to substantial performance improvements in long-horizon tasks. The proposed neural network-based solution features a simple architecture that not only facilitates ease of training and convergence but also demonstrates high efficiency and effective performance. Extensive experimental results reveal that our method exhibits strong performance across a variety of tasks.

## 1 INTRODUCTION

Offline reinforcement learning, which focuses on training agents using pre-collected static datasets without real-time environment interactions, has emerged as a promising approach, particularly in scenarios where online data collection is impractical or risky (Levine et al., 2020b). Unlike traditional reinforcement learning, where agents learn through trial and error, Offline reinforcement learning leverages offline data to teach agents step-by-step decision-making. Despite its potential, Offline reinforcement learning often encounters significant challenges related to the stability of the learning process, particularly when combined with function approximation and bootstrapping (Mazoure et al., 2023).

The primary issue arises from bootstrapping, where future estimates are used to update current predictions, in combination with function approximation, which generalizes over large state spaces (Sutton & Barto, 2018). Together, these factors propagate errors through iterative updates, which can result in divergence and lead to ineffective policy learning. The iterative propagation of these errors, especially in long-horizon tasks, causes inaccuracies to accumulate, eventually destabilizing the learning process and yielding suboptimal policies (Szepesvari & Littman, 1999).

To address these issues, recent works have proposed alternative approaches that avoid direct value estimation, opting instead for methods that focus on planning and optimizing action sequences over time (Ajay et al., 2023; Chen et al., 2021). This shift is particularly beneficial in long-horizon decision-making tasks, which mitigates the instability inherent in dynamic programming methods, which are prone to compounding approximation errors over long horizons. However, these prevailing methods are constrained by the limitations of their underlying generative models, particularly in their reliance on accurately modeling the trajectory distribution dynamics and state transitions.These models often need to capture the complexity of high-dimensional or stochastic trajectories, and their reliance on large, diverse datasets for accurate state

representation and generalization exacerbates computational cost and model complexity, particularly in data-scarce environments. (Yang et al., 2022).

Recent studies also have shown that classification-based objectives, particularly cross-entropy loss, offer a more stable alternative to regression-based losses in deep reinforcement learning, This is because cross-entropy encourages the model to learn finer distinctions and more detailed information about the data (Zhang et al., 2023). Cross-entropy loss is widely used in supervised learning tasks due to its ability to handle mitigate overfitting and noisy data issues, However, although Farebrother et al. (2024) leveraged cross-entropy loss to improve the stability of model training, it does not fundamentally address the core issues related to bootstrapping and value function estimation. Specifically, the method still relies on future estimates to update current predictions, inherently propagating the approximation errors throughout the training process.

Inspired by the approaches used in generative models, we propose a minimalistic model that transforms long-horizon planning into high-dimensional action outputs, which is called **A**ction **S**equence **P**lanner (**AS-Planner**). Our model enhances policy optimization by focusing on the trajectory level, extracting relevant patterns from sub-optimal trajectories to refine decision-making in a more structured and efficient manner. Unlike traditional generative models that often rely on complex architectures to model temporal dependencies, our approach leverages a simple neural network to implicitly model the temporal structure by directly outputting action sequences. This eliminates the need for complex trajectory modeling, making it possible for even minimal multi-layer perceptrons (MLPs) (Rumelhart et al., 1986) to handle temporal dynamics effectively. ASPlanner implicitly captures temporal dependencies without relying on value estimation and bootstrapping. By directly optimizing the policy, our method bypasses the need for future state estimations, thus reducing the error propagation that typically occurs in traditional value-based methods.

In traditional policy gradient methods (Sutton et al., 1999), Maximum Likelihood Estimation (MLE) is commonly used to maximize the likelihood of observed actions by minimizing the negative log-likelihood loss. However, MLE often suffers from high variance, particularly in reinforcement learning scenarios where step-by-step exploration can lead to unstable gradients and slow convergence (Rückstieß et al., 2008). To address these issues, ASPlanner replaces the MLE objective with a cross-entropy loss function, which compares the policy's predicted action probabilities with a target distribution derived from normalized actions. This substitution reduces the variance in gradient estimation, providing a more stable and effective approach to policy optimization in offline reinforcement learning environments. Extensive experiments showed that ASPlanner outperforms mainstream offline RL algorithms on long-horizon tasks. It significantly improves stability and policy performance in complex decision-making scenarios. Our contributions can be summarized as follows:

- We introduce a novel method that simplifies the complexity of long-horizon tasks by transforming it into an action space complexity, making it easier to handle high-dimensional action spaces without compounding errors from traditional value-based methods.

- By employing a cross-entropy loss function instead of the maximum likelihood loss, we observed significant performance improvements across various datasets.

- Our model, consisting of a few MLP layers, demonstrates strong performance on long-horizon tasks and traditional control benchmarks, providing a lightweight and efficient solution for offline reinforcement learning challenges.

## 2 PRELIMINARIES

In this section, we introduce the foundational concepts and mathematical formulations essential for understanding our proposed approach. These include the Markov Decision Process (MDP) (Bellman et al., 1957), policy gradient methods, cross-entropy loss (Shannon, 1948), and offline reinforcement learning.

## 2.1 MARKOV DECISION PROCESS

A Markov Decision Process provides a formal framework for modeling decision-making problems where outcomes are partly random and partly under the control of a decision maker. An MDP is defined by the tuple $(\mathcal{S}, \mathcal{A}, P, r, \gamma)$, where $\mathcal{S}$ is a set of states, $\mathcal{A}$ is a set of actions, $P : \mathcal{S} \times \mathcal{A} \times \mathcal{S} \to [0, 1] \subseteq \mathbb{R}$ is a state transition probability function, where $P(s_{t+1}|s_t, a_t)$ denotes the probability of transitioning from state $s_t$ to state $s_{t+1}$ in the $t$-th step after action $a_t$ is taken, $r(s_t, a_t) \in \mathbb{R}$ represents the immediate reward received after taking action $a_t$ in state $s_t$, with $\gamma \in [0, 1)$ being the discount factor that determines the importance of future rewards. The goal of reinforcement learning in an MDP is to find a policy $\pi^*(a|s)$ that maximizes the expected cumulative reward over time.

## 2.2 POLICY GRADIENT METHODS

Policy gradient methods aim to optimize the policy $\pi_\theta(a|s)$ parameterized by $\theta$ by directly following the gradient of the expected cumulative reward with respect to $\theta$. The objective function is expressed as

$$\mathcal{J}(\theta) := \mathbb{E}_{\tau \sim \pi_\theta} \left[ \sum_{t=0}^{T} \gamma^t r(s_t, a_t) \right],$$

where $\tau = (s_0, a_0, \ldots, s_T, a_T) \sim \pi_\theta$ is a trajectory sampled from $\pi_\theta$.

The policy gradient approach provides a way of computing the gradient of $\mathcal{J}(\theta)$ with respect to $\theta$:

$$\nabla_\theta \mathcal{J}(\theta) \triangleq \mathbb{E}_{\tau \sim \pi_\theta} \left[ \sum_{t=0}^{T} \nabla_\theta \log \pi_\theta(a_t|s_t) G_t \right],$$

where $G_t = \sum_{k=t}^{T} \gamma^k r(s_k, a_k)$ is the return following time step $t$. This gradient is then leveraged to perform stochastic gradient ascent on $\theta$.

## 2.3 CROSS-ENTROPY LOSS

The cross-entropy loss is widely used in classification tasks and measures the difference between the predicted distribution and the true distribution (e.g., target labels as special cases). In the context of policy learning, cross-entropy loss can be employed to align the policy's action distribution with a target distribution. For the predicted action probabilities $\hat{y}_t$ and target probabilities $y$ with $\sum_t \hat{y}_t = \sum_t y_t = 1$, the cross-entropy loss is written as

$$\ell_{\text{CE}} := - \sum_{t=0}^{T} y_t \log \hat{y}_t. \tag{1}$$

Minimizing this loss ensures that the learned policy assigns higher probabilities to actions that align with the target distribution.

## 2.4 OFFLINE REINFORCEMENT LEARNING

In offline reinforcement learning, the policy is learned from a fixed dataset of past experiences without any interaction with the environment during training (Levine et al., 2020a). Offline RL presents unique challenges, such as distributional shift between the dataset and the learned policy, making it prone to instability. Stabilizing the learning process often requires techniques to constrain the learned policy within the empirical distribution (Kumar et al., 2019), which is particularly challenging in long-horizon tasks.

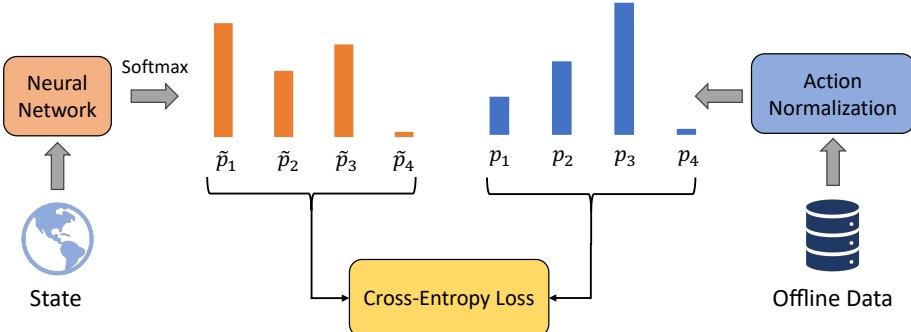

Figure 1: Cross-Entropy Loss for Policy Gradient Optimization. This figure illustrates the proposed method of replacing MLE with a CE loss function in policy gradient methods. Starting from given states, the neural network generates an action distribution through a log-softmax operation. Offline action data is normalized to form a target action distribution over the action space. The cross-entropy loss then computes the divergence between the neural network's predicted action distribution and the target distribution, guiding the learning process.

## 3 METHODOLOGY

In this section, we present our contribution, which introduces replacing the traditional maximum likelihood estimation with a cross-entropy loss function in policy gradient methods, along with the integration of action sequence planning for long-horizon tasks. The cross-entropy loss improves training stability and enhances performance. Furthermore, by enabling the policy to predict sequences of actions rather than individual actions at each time step, our method facilitates more effective planning over extended time horizons.

### 3.1 REPLACING MLE WITH CROSS-ENTROPY LOSS

To begin with, we study policy learning by optimizing action sequences. Inspired by the properties of the cross-entropy loss in equation 1, it is natural to consider guiding the learned model toward predicting action sequences $\hat{y}_t^\theta$ consistent with some target action sequences $y_t$ in the form of a distribution over a sequence of actions:

$$\ell_{\text{CE}}(\theta) := -\sum_{t=0}^{T} y_t \log \hat{y}_t^\theta.$$

Because it is commonly assumed that sampled trajectories are i.i.d., it is safe to focus on each trajectory independently and model dependencies within the trajectory sequence. In this way, we may adopt any approach that transforms the empirical action sequence $a := (a_0, \ldots, a_T)$ to a legal probability distribution $(y_0(a), \ldots, y_T(a))$ where with a little abuse of notation, we reuse $a$ for the whole action sequence when context is clear. Meanwhile, the policy model is expected to yield a distribution $(\hat{y}_0^\theta(s), \ldots, \hat{y}_T^\theta(s))$ given a start state $s$.

A key component of our method is the normalization of actions across all time steps within an episode. Instead of using raw action values, we normalize the entire sequence of actions to form a target distribution $y_t$ over the action space $\mathcal{A}$. This normalization ensures consistent scaling from the first time step 0 to the final time step $T$. For continuous action spaces, we normalize the actions at each time step $t \in [0, T]$ to form a valid probability distribution over the actions taken during the episode. The normalization we adopt is defined as $y_t(a) = \frac{a_t - a_{\min}}{\sum_{t'=0}^{T}(a_{t'} - a_{\min})}$, where $a_{\min} := \min_t a_t$ is the minimum action value among all the

actions in the trajectory. This ensures that $y_t \geq 0$ and that the sum over all time steps satisfies $\sum_{t=0}^{T} y_t = 1$. By forming this target distribution over the actions taken in the episode, the policy is able to learn from the entire sequence of actions proportionally, rather than focusing solely on individual actions.

In traditional policy gradient methods, MLE is used to maximize the likelihood of observed actions by minimizing the negative log-likelihood loss. In our approach, we replace the MLE objective with a cross-entropy loss function that compares the policy's action sequence distribution with the target distribution formed by the normalized actions. The weighted cross-entropy loss for policy learning is defined as

$$\ell_{\text{CE}}(\theta) := -\sum_{t=0}^{T} y_t(a) G_t \log \bar{\pi}_{t;\theta}(\hat{a}),$$

where $\theta$ denotes the policy parameters, $G_t = \sum_{k=t}^{T} \gamma^k r_k$ is the discounted return from time step $t$, with $\gamma$ being the discount factor and $r_k$ being the reward at time $k$. $\bar{\pi}_{t;\theta}(\hat{a})$ normalizes the expected action $\hat{a}_t = \mathbb{E}_{a_t} \pi_\theta(a_t|s_t) a_t$ over $0 \leq t \leq T$, under the policy $\pi_\theta$, and $y_t(a)$ is the normalized weight for action $a_t$ at time $t$.

This loss function encourages the policy to produce action distributions that align with the target distribution $y_t$, leading to more stable and consistent policy updates. By incorporating the discounted returns $G_t$ into the weighting, we focus the learning process on actions that yield higher returns. For a detailed explanation of the code implementation and further experimental validation of this approach, we refer the reader to subsection A.1, which provides the implementation of both the classical policy gradient and our modified approach.

Additionally, we further improve the stability and performance of the policy updates by incorporating the Kullback-Leibler (KL) divergence (Kullback & Leibler, 1951) between the target action distribution $y_t(a)$ and the policy distribution $\pi_\theta(a|s_t)$. The KL divergence between $y_t(a_t)$ and $\pi_\theta(a_t|s_t)$ is expressed as

$$\text{KL}(y_t(a_t)||\pi_\theta(a_t|s_t)) = \sum_{t=0}^{T} y_t(a_t) \left( \log y_t(a_t) - \log \pi_\theta(a_t|s_t) \right),$$

where we take advantage of the original likelihood $\pi_\theta(a_t|s_t)$ as in vanilla policy gradient methods.

By minimizing this KL divergence, the policy distribution $\pi_\theta(\cdot|s_t)$ is encouraged to align more closely with the target distribution $y_t(a)$. This provides a more stable training signal, as it penalizes large discrepancies between the predicted and target distributions. The overall loss function of the model is now composed of the cross-entropy loss, augmented by an additional regularization term based on the KL divergence. The final loss function $L_{\text{total}}(\theta)$ can be written as

$$\mathcal{L}_{\text{total}}(\theta) = \mathcal{L}_{\text{CE}}(\theta) + \lambda \cdot \mathcal{L}_{\text{KL}}(\theta).$$

Here, $\mathcal{L}_{\text{CE}}(\theta)$ is the cross-entropy loss as defined earlier, and the term $\lambda$ is a weighting factor that controls the influence of the KL divergence regularization. This hyperparameter $\lambda$ can be tuned to balance the trade-off between fitting the cross-entropy loss and maintaining a close resemblance between the policy and target distributions.

## 3.2 ACTION SEQUENCE PLANNING FOR LONG-HORIZON TASKS

To effectively implement the action sequence planning mechanism, our key modification lies in the design of the neural network's output layer. Specifically, instead of outputting a single action per time step, the model generates a sequence of actions by setting the output layer's dimensionality to $2 \times k \times$ action_dim, where $k$ represents the length of the action window and action_dim is the dimensionality of each individual action. The factor of 2 arises from the need to parameterize a Gaussian distribution for each action in the

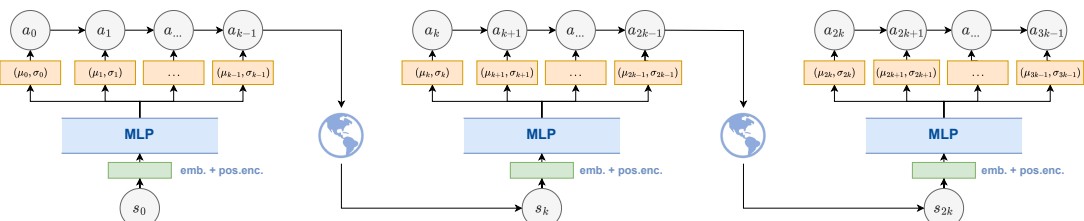

Figure 2: The overall model architecture for action sequence planning in long-horizon tasks. Starting with an initial state $s_0$, the input is passed through an embedding layer and position encoding before being processed by an MLP, which outputs $\mu$ and $\sigma$ parameters for a Gaussian distribution over the next $k$ actions. After executing these actions, the next state $s_k$ is used to predict the subsequent $k$ actions, continuing the process.

sequence. For each action $a_t$, the policy network outputs both the mean $\mu_t$ and the standard deviation $\sigma_t$, which together define a Gaussian distribution $\mathcal{N}(\mu_t, \sigma_t^2)$. The total output of the network at each time step is $(\mu_t, \sigma_t, \mu_{t+1}, \sigma_{t+1}, \ldots, \mu_{t+k-1}, \sigma_{t+k-1})$. By predicting both the mean and the standard deviation, the policy is able to sample actions from the Gaussian distribution during execution.

It should be noted that when the value of $k$ is set to the maximum sequence length, the model essentially performs a one-step planning mechanism, outputting the entire sequence of actions based on the initial state at the first time step. This approach has proven to be highly effective in tasks where randomness plays a minimal role. Conversely, when $k = 1$, the model generates only a single action at each time step, closely resembling traditional policy gradient algorithms. This setting grants the algorithm greater adaptive flexibility but sacrifices the ability to plan future actions in advance. Thus, selecting an appropriate value for $k$ allows us to strike a balance between planning capacity and adaptive adjustment. Depending on the specific task, different window sizes can be explored to optimize the trade-off between these two aspects.

This output structure not only facilitates long-term planning but also integrates uncertainty into the decision-making process, making the model more robust in complex environments where the state-action mapping may be non-deterministic. The training of this model follows the standard policy gradient framework, but, as discussed earlier, the maximum likelihood estimation is replaced by cross-entropy loss weighted by the return $G_t$. The policy update involves minimizing this cross-entropy loss and KL divergence , ensuring that the predicted action distribution aligns with the target distribution over the sequence of actions. The training procedure is outlined in Algorithm 1

---

**Algorithm 1** Training Procedure with Cross-Entropy Loss

---

Initialize network parameters $\theta$, sequence length $T$ and window size $k$
**for** each episode **do**
    Sample initial state $s_0$
    **for** each time step $t = 0, k, 2k, \ldots, T-k$ **do**
        Output action sequence $(\mu_{t:t+k-1}, \sigma_{t:t+k-1})$ for the next $k$ steps based on state $s_t$
        Sample actions $a_{t:t+k-1}$ from $\mathcal{N}(\mu_{t:t+k-1}, \sigma_{t:t+k-1})$
    **end for**
    Normalize actions: $y_t(a) = \frac{a_t - a_{\min}}{\sum_{t=0}^{T}(a_t - a_{\min})}$
    Update policy parameters: $\theta \leftarrow \theta - \alpha \nabla_\theta \mathcal{L}_{\text{total}}(\theta)$
**end for**

---

# 4 EXPERIMENTS

To validate the efficacy of ASPlanner, we conducted experiments across multiple environments, including D4RL and Budget Allocation Environment. This section details the experimental setups, results, and insights gained from these evaluations.

## 4.1 D4RL ENVIRONMENT

Table 1: These scores represent the return obtained from executing a policy in the D4RL simulator, averaged over 3 seeds. Specifically, we report results under three different configurations of our proposed method. The first is **ASPlanner (best score)**, which represents the performance of our approach with optimal parameter tuning. The second is **ASPlanner ($k = 1$)**, where the window length $k$ is fixed to 1 while retaining the other parameters from the best configuration. Finally, the third is **ASP-MLE**, which substitutes the CE loss in our method with the MLE loss, keeping all other settings the same as in the optimal configuration.

| | SAC | BC | SAC-off | BEAR | BRAC-p | BRAC-v | AWR | BCQ | aDICE | CQL | ASP-MLE | ASPlanner(k=1) | ASPlanner(best score) |
|---|---|---|---|---|---|---|---|---|---|---|---|---|---|
| maze2d-umaze | 110.4 | 29.0 | **145.6** | 28.6 | 30.4 | 1.7 | 25.2 | 41.5 | 2.2 | 31.7 | 39.6 | 42.1 | 101.3 |
| maze2d-medium | 69.5 | 93.2 | 82.0 | 89.8 | 98.8 | 102.4 | 33.2 | 35.0 | 39.6 | 26.4 | 43.4 | 50.2 | **131.2** |
| maze2d-large | 14.1 | 20.1 | 1.5 | 19.0 | 34.5 | 115.2 | 70.1 | 23.2 | 6.5 | 40.0 | 12.5 | 11.2 | **152.7** |
| hammer-human | -248.7 | -82.4 | -214.2 | -242.0 | -239.7 | -243.8 | -115.3 | -210.5 | -234.8 | **300.2** | -298.8 | -98.7 | -40.9 |
| door-human | -61.8 | -41.7 | 57.2 | -66.4 | -66.5 | -66.4 | -44.4 | -56.6 | -56.5 | **234.3** | -63.7 | -12.1 | -1.7 |
| relocate-human | -13.7 | -5.6 | -4.5 | -18.9 | -19.7 | -19.7 | -7.2 | -8.6 | -10.8 | 2.0 | -9.3 | -7.6 | **10.3** |
| hammer-cloned | -248.7 | -175.1 | -244.1 | -241.1 | -236.7 | -236.9 | -226.9 | -224.4 | -233.1 | **-0.41** | -276.1 | -235.1 | -223.2 |
| door-cloned | -61.8 | -60.7 | -56.3 | -60.9 | -58.7 | -59.0 | -56.1 | -56.3 | -56.4 | **-44.76** | -62.0 | -56.2 | -54.9 |
| relocate-cloned | -13.7 | -10.1 | -16.1 | -17.6 | -19.8 | -19.4 | -16.6 | -17.5 | -18.8 | -10.66 | -11.3 | -56.4 | **1.2** |
| hammer-expert | -248.7 | 16140.8 | 3019.5 | 6359.7 | -241.4 | -241.1 | 4822.9 | 13731.5 | -235.2 | 11062.4 | -223.4 | 2521 | **16211.2** |
| door-expert | -61.8 | 969.4 | 163.8 | 2980.1 | -66.4 | -66.6 | 2964.5 | 2850.7 | -56.5 | -66.7 | 2926.8 | 1821.1 | **2983.1** |
| relocate-expert | -13.7 | 4289.3 | -18.2 | 4173.8 | -20.6 | -21.4 | 3875.5 | 1759.6 | -8.7 | 4019.9 | -7.1 | 241.9 | **4302.4** |

We performed extensive offline learning experiments using the D4RL benchmark suite. The performance of these algorithm is reported in Fu et al. (2021). Our algorithm shows significant improvement in maze2d-medium and maze2d-large tasks. The performance on these long-horizon tasks demonstrates the strength of ASPlanner's ability to handle extended sequences of decisions, suggesting robust planning capabilities. In the Adroit domain, which requires high precision and control, ASPlanner also excelled in several tasks. Our algorithm surpassed many methods by a significant margin, highlighting its proficiency in mastering fine-grained manipulation tasks.

## 4.2 BUDGET ALLOCATION ENVIRONMENT

As described in subsection A.2, We set up a Budget Allocation Environment, where the objective is to allocate a fixed budget optimally across different time steps over a given time horizon. In this environment, as shown in the figure, $T$ represents the sequence length, and as $T$ increases, the difficulty of the task rises significantly. This is because it becomes more challenging to decide how to allocate the budget effectively across more time steps while ensuring optimal returns.

To evaluate performance, we set the score from uniformly allocating the budget as the baseline 0, and the score from the theoretical optimal solution as 1. The normalized score for each method is calculated using the formula: Normalized Score $= \frac{S_{\text{alg}} - S_{\text{uniform}}}{S_{\text{opt}} - S_{\text{uniform}}}$ where $S_{\text{alg}}$ is the method's score, $S_{\text{uniform}}$ is the score for uniform allocation, and $S_{\text{opt}}$ is the score for the optimal solution. This allows for precise comparison, including methods that perform worse than the baseline.

We evaluate our algorithm in this environment, comparing it to the baselines **BCQ** (Fujimoto et al., 2018), **CQL** (Kumar et al., 2020), **IQL** (Kostrikov et al., 2022), and **TDMPC2** (Hansen et al., 2024). Our method consistently outperforms these algorithms, especially as the sequence length $T$ increases. Unlike traditional offline RL methods, which often suffer from error propagation and instability due to bootstrapping and

Table 2: These scores represent the return obtained from executing a policy in the Budget Allocation Environment, averaged over 3 seeds. Specifically, we report results under three different configurations of our proposed method. The first is **ASPlanner (best score)**, which represents the performance of our approach with optimal parameter tuning. The second is **ASPlanner** ($k = 1$), where the window length $k$ is fixed to 1 while retaining the other parameters from the best configuration. Finally, the third is **ASP-MLE**, which substitutes the CE loss in our method with the MLE loss, keeping all other settings the same as in the optimal configuration.

|  | BCQ | CQL | IQL | TDMPC2 | ASP-MLE | ASPlanner(k=1) | ASPlanner(best score) |
|---|---|---|---|---|---|---|---|
| $T = 10$ | -0.73 | -0.68 | 0.07 | -0.66 | 0.85 | 0.85 | **0.97** |
| $T = 20$ | -0.58 | -0.50 | -0.25 | -0.49 | 0.80 | 0.80 | **0.91** |
| $T = 30$ | -0.61 | -0.91 | -0.03 | -0.91 | 0.17 | 0.16 | **0.86** |
| $T = 50$ | 0.08 | -0.25 | 0.13 | -0.25 | 0.11 | 0.09 | **0.85** |
| $T = 100$ | -0.22 | -0.32 | -0.20 | -0.32 | 0.09 | 0.08 | **0.81** |

value estimation, our approach remains robust. By planning the entire action sequence at the initial step, we reduce variance and maintain high performance, even as the difficulty of budget allocation increases with longer horizons.

### 4.3 COMPARISON OF CROSS-ENTROPY LOSS AND LIKELIHOOD LOSS

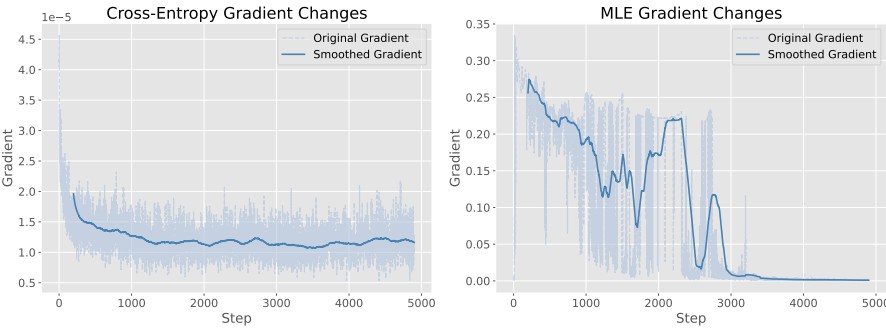

Figure 3: Gradient behavior comparison between Cross-Entropy Loss and Likelihood Loss. Cross-Entropy Loss shows smoother gradients during updates, while Likelihood Loss exhibits more variability.

We compared the cross-entropy loss function with the traditional maximum likelihood estimation in the environment. Cross-entropy loss exhibited smoother gradient updates, leading to reduced training noise and improved stability. In contrast, MLE resulted in noisier gradients, often causing instability. Additionally, models trained with cross-entropy loss consistently achieved faster convergence and higher final performance scores compared to MLE. This demonstrates the effectiveness of cross-entropy loss in enhancing both training stability and model performance.

In addition to analyzing gradient behavior, we compared the performance of CE loss and MLE loss across various tasks in our experimental setups. During these experiments, we first tuned the action window length $k$ and other relevant hyperparameters to their optimal values for each task. Once these parameters were fixed, we replaced the loss function with MLE, referred to as ASP-MLE, to evaluate the impact of different loss functions under the same conditions. As demonstrated in Tables 1 and 2, the use of CE loss consistently

outperformed MLE in terms of task performance, the Budget Allocation environment further corroborates these findings. As shown in Budget Allocation environment, CE loss consistently resulted in higher returns as the episode length increased, this stark difference illustrates that CE loss not only smoothens the gradient updates, as depicted in Figure 3, but also leads to more effective policy optimization, particularly in environments where planning over long time horizons is critical.

## 5 RELATED WORK

Offline reinforcement learning has emerged as a powerful tool for learning policies from pre-collected datasets, particularly when real-time interactions with the environment are costly or unsafe. However, one of the key challenges in offline RL is handling instability caused by bootstrapping and function approximation, especially in long-horizon tasks.

**Bootstrapping and Function Approximation:** Bootstrapping can lead to instability by propagating estimation errors through the learning process. To address this, Wang et al. (2022) developed the Bootstrapped and Constrained Pessimistic Value Iteration (BCP-VI) algorithm, which utilizes bootstrapping and pessimism to reduce suboptimality in offline RL with linear function approximation. Bai et al. (2022) proposed Pessimistic Bootstrapping for Offline RL (PBRL), which penalizes out-of-distribution (OOD) actions to mitigate extrapolation error while generalizing beyond the offline data.

**Generative Modeling and Policy Optimization:** Generative models have also been leveraged to tackle offline RL problems. Wei et al. (2021) developed Action-conditioned Q-learning (AQL), which incorporates generative modeling to improve policy approximation by mitigating distribution shift. Yin et al. (2023) further explored differentiable function approximation in offline RL, providing a theoretical framework for understanding the statistical complexity of non-linear function approximators.

**Cross-Entropy Methods for Stability:** Cross-entropy loss has been increasingly used in RL to address instability in training. The work by Wen & Topcu (2020) introduced a constrained cross-entropy method for safe reinforcement learning, ensuring stability by transforming constrained optimization problems into unconstrained ones.

**Robustness in Offline RL:** Addressing the issue of out-of-distribution actions and policy divergence, Bai et al. (2024) proposed a Monotonic Quantile Network (MQN), which improves the robustness of policy learning by optimizing a worst-case criterion of returns. This method is especially effective in safety-critical applications where avoiding risky actions is paramount.

## 6 CONCLUSION

This paper presented a novel approach to offline reinforcement learning that replaces maximum likelihood estimation with a cross-entropy loss function to improve training stability and performance in long-horizon tasks. Our method, ASPlanner, generates entire action sequences, reducing error propagation and enhancing decision-making by avoiding step-by-step planning.

Experiments on D4RL and Budget Allocation Environments showed that ASPlanner performs strongly across many tasks, with faster convergence and improved stability compared to existing methods. By directly optimizing action sequences and leveraging cross-entropy loss, our approach effectively handles the complexity of high-dimensional action spaces and long-term planning.

Given that our method relies on a simple neural network architecture with only a few layers, future work can build on this minimalistic learning framework. We hope our work inspires further research and applications in both academia and industry, contributing to the development of more efficient offline reinforcement learning methods.

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

# A APPENDIX

## A.1 SPECIFIC IMPLEMENTATION CODE

Our method replaces the MLE objective with a cross-entropy loss and outputs action sequence, which allows the model to predict a sequence of actions rather than a single action at each time step. We now provide a code-based implementation of our method, contrasting it with the classical policy gradient approach.

**Classical Policy Gradient:**

In classical policy gradient methods, the policy is updated by maximizing the action likelihood using MLE, outputting the mean and standard deviation for a single time step.

Listing 1: Classic Policy Gradient

```
#----------------Classic Policy Gradient-------------
class Neural_Network(nn.Module):
    def __init__(self, state_dim, hidden_size, output_size):
        #output_size = 2 * action_dim
        super(Neural_Network, self).__init__()
        self.relu = nn.ReLU()
        self.fc1 = nn.Linear(state_dim, hidden_size)
        self.fc2 = nn.Linear(hidden_size, hidden_size)
        self.fc3 = nn.Linear(hidden_size, output_size)

    def forward(self, x):
        out = self.fc1(x)
        out = self.relu(out)
        out = self.fc2(out)
        out = self.relu(out)
        out = self.fc3(out)
        return out

def update_pi(self, z, action_list, reward_list):
    # z: state embedding with position encoding, shape (epi_len, batch_size,
        embedding_dim)
    # action_list: offline action data, shape (epi_len, batch_size, action_dim)
    # reward_list: offline reward data, shape (epi_len, batch_size)
    self.pi_optim.zero_grad(set_to_none=True)
    mean, std = self.model.pi(z, action_list)

    log_probs = []
    discounted_rewards = np.zeros_like(reward_list.cpu().detach().numpy())
    running_add = 0
    gamma = 0.999

    for t in reversed(range(len(reward_list))):
        dist = torch.distributions.Normal(mean[t], std[t])
        actions = action_list[t]
        log_prob = dist.log_prob(actions.to(device=self.device)).to(device=self
            .device)

        log_probs.append(log_prob)
        running_add = running_add * gamma + reward_list[t]
        discounted_rewards[t] = running_add.cpu().detach().numpy()
```

```
40
41     log_probs = torch.stack(log_probs)
42     discounted_rewards = torch.tensor(discounted_rewards, dtype=torch.float32).
           to(device=self.device)
43
44     discounted_rewards_mean = discounted_rewards.squeeze(-1).mean(dim=1,
           keepdim=True)
45     discounted_rewards_std = discounted_rewards.squeeze(-1).std(dim=1, keepdim=
           True) + 1e-5
46     discounted_rewards = (discounted_rewards.clone().squeeze(-1) -
           discounted_rewards_mean) / discounted_rewards_std
47
48     pi_loss = - (log_probs * discounted_rewards.unsqueeze(-1)).mean()
49     pi_loss.backward()
50     self.pi_optim.step()
51
52     return pi_loss
```

**Our Method:**

In our method, we replace the MLE loss with a cross-entropy loss that incorporates normalized action distributions and modify the neural network to output k actions at each step. Below is the implementation of our policy update step:

Listing 2: Our Method

```
1  #----------------Our Method------------
2  class Neural_Network(nn.Module):
3      def __init__(self, state_dim, hidden_size, output_size):
4          #output_size= 2 * k * action_dim
5          super(Neural_Network, self).__init__()
6          self.relu = nn.ReLU()
7          self.fc1 = nn.Linear(state_dim, hidden_size)
8          self.fc2 = nn.Linear(hidden_size, hidden_size)
9          self.fc3 = nn.Linear(hidden_size, output_size)
10
11     def forward(self, x):
12         out = self.fc1(x)
13         out = self.relu(out)
14         out = self.fc2(out)
15         out = self.relu(out)
16         out = self.fc3(out)
17         return out
18
19 def update_pi(self, z, action_list, reward_list, update_times=0):
20     # z: state embedding with position encoding, shape (epi_len, batch_size,
           embedding_dim)
21     # action_list: offline action data, shape (epi_len, batch_size, action_dim)
22     # reward_list: offline reward data, shape (epi_len, batch_size)
23     self.pi_optim.zero_grad(set_to_none=True)
24     mean, std = self.model.pi(z, action_list, update_times=update_times)
25
26     pred = F.log_softmax(mean.clone().permute(1, 0, 2), dim=1)
27     target = action_list.clone().permute(1, 0, 2).to(device=self.device)
28     target_shifted = target - self.cfg.action_min
29     target_sum = target_shifted.sum(dim=1, keepdim=True)
```

```
30      target = target / (target_sum + 1e-5)
31      loss_distribution = ((target * pred).permute(1, 0, 2)).to(device=self.
            device)
32
33      discounted_rewards = np.zeros_like(reward_list.cpu().detach().numpy())
34      running_add = 0
35      gamma = 0.999
36
37      for t in reversed(range(len(reward_list))):
38          running_add = running_add * gamma + reward_list[t]
39          discounted_rewards[t] = running_add.cpu().detach().numpy()
40
41      offline_mean = action_list.mean(dim=1)
42      offline_std = action_list.std(dim=1) + 1e-5
43      offline_dist = torch.distributions.Normal(offline_mean, offline_std)
44
45      kl_divergence = (torch.distributions.kl.kl_divergence(torch.distributions.
            Normal(mean.mean(dim=1), std.mean(dim=1)),
46                                          offline_dist)).mean()
                                                .to(device=self.
                                                device)
47
48      discounted_rewards = torch.tensor(discounted_rewards, dtype=torch.float32).
            to(device=self.device)
49      discounted_rewards_mean = discounted_rewards.squeeze(-1).mean(dim=1,
            keepdim=True)
50      discounted_rewards_std = discounted_rewards.squeeze(-1).std(dim=1, keepdim=
            True) + 1e-5
51      discounted_rewards = (discounted_rewards.clone().squeeze(-1) -
            discounted_rewards_mean) / discounted_rewards_std
52
53      pi_loss = (- (1 * loss_distribution * discounted_rewards.unsqueeze(-1)).
            mean() + self.cfg.kl_para * kl_divergence)
54
55      pi_loss.backward()
56      self.pi_optim.step()
57
58      return pi_loss
```

Intuitively, the essence of MLE lies in evaluating the similarity between the actions predicted by the neural network and the offline actions recorded in the dataset. The more similar these actions are, the smaller the loss value becomes. Similarly, the CE loss function measures the divergence between the predicted and target action distributions. By minimizing this divergence, we achieve a direct measure of how well the model's predictions align with the offline data. From this perspective, the CE loss can be viewed as a natural extension of MLE, both conceptually and mathematically.

Building on this intuition, our method replaces the traditional MLE loss in policy gradient optimization with the CE loss. This substitution is empirically validated through extensive experimentation. Our results demonstrate that this approach enhances the stability of training and improves the overall performance of the policy, particularly in long-horizon tasks.

Moreover, the addition of KL divergence regularization further strengthens the stability of policy updates by penalizing significant deviations between the predicted and target distributions, which is critical for maintaining robustness during training.

## A.2 BUDGET ALLOCATION ENVIRONMENT

In this appendix, we formally describe the reinforcement learning environment used in this work, which we refer to as the *Budget Allocation Environment*. The state at time $t$, denoted as $S_t$, is defined as a tuple of three elements:

$$S_t = (\text{timeLeft}_t, \text{bgtLeft}_t, \text{bgtCost}_t)$$

$\text{timeLeft}_t = \frac{t}{T} \in [0, 1]$ represents the normalized remaining time at time step $t$, where $T$ is the total number of decision steps in the environment. $\text{bgtLeft}_t = \frac{\text{remainingBudget}}{\text{budget}} \in [0, 1]$ represents the proportion of the remaining budget at time $t$. $\text{bgtCost}_t = A_{t-1} \in [0, 1]$ records the proportion of the budget spent at the previous time step $t - 1$, and it is equal to the action taken at that step.

At each time step, the agent chooses an action $A_t \in [0, 1]$, representing the proportion of the total budget to allocate at time step $t$. The environment provides a reward $R_t$, where $R_t \in [0, \infty]$, reflecting the value gained by allocating that portion of the budget.

The state transitions are governed by the following dynamics:

$$P(S_{t+1} \mid S_t, A_t)$$

Specifically, the next state $S_{t+1}$ is determined by:

$$\text{timeLeft}_{t+1} = t + \frac{1}{T}$$

$$\text{bgtLeft}_{t+1} = \text{bgtLeft}_t - A_t$$

$$\text{bgtCost}_{t+1} = A_t$$

Thus, the remaining time and remaining budget are updated based on the action taken at each time step, while $\text{bgtCost}_{t+1}$ records the action taken at the current time step, to be used in the next time step.

The reward function $R(S_t, A_t)$ is defined as:

$$R(S_t, A_t) = \alpha_t (A_t - \beta_t)^2$$

where $\alpha_t$ is a time-dependent scaling factor, and $\beta_t$ represents the optimal budget allocation proportion for time step $t$. This quadratic reward structure indicates that the reward increases as the budget allocation $A_t$ approaches $\beta_t$, though the reward growth diminishes as the allocation deviates from this ideal.

The purpose of the *Budget Allocation Environment* is to simulate various budget spending scenarios and to evaluate different allocation strategies over a fixed time horizon. By defining different reward functions $f(s_t, a_t)$ and transition mechanisms $\mathcal{T}(s_t, a_t)$, the environment can reflect diverse real-world spending conditions and allow for the calculation of optimal strategies. The environment's multi-dimensional and non-linear characteristics help isolate factors that might otherwise confound real-world analysis.

