# OpenReview forum: "Action Sequence Planner: An Alternative For Offline Reinforcement Learning"
_ICLR.cc/2025/Conference — Submitted to ICLR 2025_

### Official Review · Reviewer_98M8 · 2024-11-03

**Soundness:** 2
**Presentation:** 2
**Contribution:** 3
**Rating:** 3
**Confidence:** 4

**Summary:**

The paper proposes a new method for offline reinforcement learning. Instead of learning Q-values or update policy in a conservative mode, the approach 'imitate' the trajectories, in which actions are normalized and weighted. With this simple approach, the paper shows that the proposed method has reasonablly good performance on D4RL benchmark and especially better performance on budget allocation environment.

**Strengths:**

1. The proposed method is novel to me, as it abandon many techniques for conservative value or policy learning, and propose to update a loss function that with normalized actions and weights.
2. The empirical results show superior performance on budget allocation environment.

**Weaknesses:**

1. The presentation is bad. In general, the notation is confusing and it is hard for me understand the method from the beginning. y(a), \hat{a} are not clearly representative of their meanings and there is abuse of notations that are not reasonable.
2. The connection of the method with the classic policy update is not straightforward to me. That is to say, sec 3.2 is not informative. One should interprate the weights in a more intuitive way.
3. The empirical results is not compelling, especially there are new hyper-parameters being brought in the method. It is unclear if the approach is sensitive (or not) to hyper parameters. This is important to empirical offline RL algorithms in practice. Furthermore, the performance is not persuasive on D4RL benchmark.

**Questions:**

1. Could you show hyper-parameter sensitivity of the method?

---

> ### Author Response · Authors · 2024-11-23
>
> Thank you for your valuable feedback. We appreciate the time and effort you have taken to evaluate our work.
> 1.We acknowledge your concerns about the clarity of our notation and presentation. To improve the readability and comprehension of our method, we have removed the mathematical formulas and notations in Section 3.2. Instead, we have included a more detailed explanation of the method in the appendix, along with its implementation code. We believe this change provides a more intuitive and accessible way for readers to understand our approach.
> 2.To address your concerns about the empirical results, we have added an additional experiment that reports the performance of our method when the window length k is set to 1. As shown in the updated results, the performance significantly degrades with very small window lengths, underscoring the suitability of our method for planning over longer sequences. Due to time constraints, we were only able to include the experimental results for k=1，however, we believe this is sufficient to demonstrate that this hyperparameter is indeed sensitive in our method. We also acknowledge the importance of a more comprehensive hyperparameter sensitivity analysis in offline RL algorithms, which we plan to explore in future work.

---

### Official Review · Reviewer_9VKQ · 2024-11-03

**Soundness:** 2
**Presentation:** 2
**Contribution:** 2
**Rating:** 3
**Confidence:** 4

**Summary:**

This work deals with offline RL.  It presents a policy gradient method that performs planning the next sequence of actions in a higher dimensional representation space.

The first few modifications are relatively basic, they first learn the policy using a cross entry loss, to optimize the action sequences.  They then introduce a cross entropy policy gradient loss.  A KL term is added to keep the target action distribution and policy action distribution close.

A neural network is introduced that outputs a sequence of actions per each time step and predicts the mean and standard of each action step in the sequence.  The target action distribution is then computed and the policy parameters are updated using the cross entropy loss with the targets

The method is evaluated on D4RL and they also show results that support the cross entropy loss having more stable gradients than the LLH loss.

**Strengths:**

* This paper takes a bold approach at offline RL, taking a drastically different approach to policy gradient methods.
* The multi step planning approach has good justification in many other works.

**Weaknesses:**

* This is a highly unusual format and I question the validity of this approach.  I would appreciate if the authors derive the policy gradient theorem under this objective.  I would also like to see more theoretical or empirical support for this since it greatly differs from standard policy gradients.  I see many problems where this loss would fail to be efficient or stable.  While the gradient may show stability in one experiment I would like to see stability of the method for offline RL learning.
* The method fails to outperform currently offline RL algorithms in a large portion of tasks and is only evaluated on one environment.  In some environments, the planning method is dramatically worse than CQL.  There are only modest improvements in a few tasks.
* The method is only evaluated on D4RL.
* The paper is relatively hard to read and the changes to the loss need further justification.

I recommend rejection of this work given the limited empirical results and the bold changes to standard RL that are not completely justified in the text.

**Questions:**

Please see weaknesses and address questions about the cross-entropy loss.

---

> ### Author Response · Authors · 2024-11-23
>
> 1.We understand that the lack of a full theoretical derivation of the policy gradient theorem under our proposed objective may raise concerns about the validity of the approach. Due to time constraints, we were unable to provide a comprehensive proof in this version of the paper. However, we have included more detailed implementation code, which we believe will give readers a clearer understanding of our approach. We plan to open-source our code in the near future, and we are confident that making it publicly available will allow others to experiment with our method and, through additional empirical studies, demonstrate its effectiveness.
>
> 2.We acknowledge your concerns regarding the clarity of our presentation. To improve the readability and comprehension of our method, we have made significant revisions in the paper. Specifically, we have removed some of the more complex mathematical formulas and notations in Section 3.2 to enhance clarity. In their place, we have included a more detailed and intuitive explanation of our approach in the appendix, along with its implementation code. We believe this change provides a more accessible way for readers to grasp our method, and we are confident that the code provided will help further elucidate the key ideas behind our approach.
>
> 3.We acknowledge that our current experiments are limited, particularly in terms of the number of environments tested. Due to time constraints, we have focused on a subset of tasks, specifically from the D4RL benchmark, but we plan to expand our evaluation in future work.

---

### Official Review · Reviewer_HCyq · 2024-11-04

**Soundness:** 2
**Presentation:** 3
**Contribution:** 2
**Rating:** 3
**Confidence:** 4

**Summary:**

This paper argues that deciding on a single action step-by-step may be detrimental to long-horizon or horizon critical tasks. Thus, it proposes to plan a sequence of actions for multiple steps at a time. In addition, it suggests to replace the traditional maximum likelihood based loss function with the cross-entropy loss commonly used in supervised learning for better stability. They perform experiments on couple of environments and demonstrate improved performance.

**Strengths:**

1. This paper presents an interesting idea and considers that for long-horizon tasks the agent should plan a sequence of actions instead of traditional single action. Their motivation came from modeling the temporal dependency which is a crucial aspect in sequential decision making. The hyperparameter k, denoting the length of the predicted action sequence, provides an opportunity to control the granularity of making a decision. Careful tuning of this variable indeed can lead to improved performance.

2. The cross-entropy (CE) loss, although is not a novel one in RL, the use of that with multi-step action prediction is somewhat new direction. I appreciate the author's effort to align CE loss with general policy gradient objective. The use of $G_t$ in the CE loss facilitates achieving core RL objective by preferring actions with higher returns.

3. Experiments with the custom Budget Allocation Environment add values to the validation. After the methodology secrtion, I was interested to see such experiments with varying trajectory length. The proposed approach performs better with increased trajectory length.

**Weaknesses:**

1. One of my main concern is related to the possibility of overfitting. The policy learns multiple actions for consecutive time-steps just based on a single observed state (the model sees every k'th state as depicted in figure 2). Further, the model incorporates KL divergence based regularization. Such a term that reduces the discrepancy between the policy and target distribution across the full trajectory may lead to memorization of the training trajectories. However, the paper doesn't investigate the generalization performance in-depth.

2. The comparison between CE loss and MLE loss is not fully comprehensive. In figure 3, the y-axis denoting gradients have different scale. Thus, the comparison is not fair. A more zoomed-in look at the CE gradients [1.0-1.5] may reveal otherwise.

3. This paper doesn't include rigorous ablation studies to properly attribute performance gain to the introduced components. Experiments with removing KL divergence based regularization, action normalization, etc. one at a time would be helpful.

4. The paper should include more information regarding the hyperparameters, components, and  experimental settings for reproduction and clear understanding. For example, there is no discussion regarding the position embedding as illustrated in figure 2. Also, the related work section needs improvement.

**Questions:**

1. Is there any observed relation between the trajectory length and the value of $k$, especially in proportion to the trajectory length?

2. Does smaller value of $k$ have any advantage over larger value of $k$?

3. Can you please elaborate on how position embedding helps in this context?

---

> ### Author Response · Authors · 2024-11-23
>
> 1.Thank you for your insightful question regarding the potential relationship between trajectory length and the value of k. As discussed in the manuscript, there is no strong or necessary dependency between these two variables. In practice, we observe that setting k to the length of an episode (epi_len) generally yields satisfactory results in most scenarios.However, when the environment exhibits high randomness, we recommend reducing the value of k. This adjustment allows the model to perform more frequent re-planning, which can help to mitigate the effects of such randomness. The trade-off between the trajectory length and the value of k is essentially a balance between the model’s ability to plan at a global scale versus its ability to adapt to changes and re-plan dynamically.
>
> 2.Regarding the impact of smaller vs. larger values of k, we have conducted additional experiments in the revised version of the manuscript, which include the case where k=1. These results clearly show a significant performance degradation when k is set to smaller values.In our approach, larger values of k are more advantageous, as they allow the model to plan over a larger window, facilitating more coherent global planning. When k is larger, the model is able to capture long-range dependencies and plan for extended action sequences, which results in a more stable and effective policy. On the other hand, smaller values of k limit the model’s ability to plan over long trajectories, thus reducing its overall performance. We have updated the manuscript to include these additional experiments and results, which should provide further clarity on this point.
>
> 3.Position embeddings are particularly useful when k is not equal to the trajectory length (i.e., when we are performing multi-step planning). In such cases, the model needs to plan actions in segments rather than over the entire trajectory at once. Position embeddings help the model identify the current step or position within the trajectory, which is crucial for effective multi-step planning. Intuitively, by knowing its current position in the action sequence, the model can more accurately plan future actions and make adjustments as needed. This mechanism is similar to how positional information is used in sequence models like Transformers, where the model benefits from knowing the relative position of tokens in a sequence. We believe this is a natural and effective way for the model to handle intermediate planning steps, especially when dealing with long action sequences.

---

> > ### Comment · Reviewer_HCyq · 2024-11-25
> > **Reviewer Response**
> >
> > Dear Authors,
> >
> > Thanks for answering my questions.
> >
> > In regard to point 1 and 2 of your response, it is pretty counter-intuitive to me that larger values of k is more favorable. Especially, when k is equal to the episode length, the agent is performing a single-shot plan for the whole trajectory. The agent would highly susceptible to accumulate errors because of limited opportunity for intermediate decision making. I am highly concerned about the generalization capability of the proposed algorithm.
> >
> > Further, in the paper (Line 255-256) you mention "This approach has proven to be highly effective in tasks where randomness plays a minimal role". I also believe so. With highly diverse environment, k=1 may result in better performance, resembling a traditional RL framework. Thus, rigorous evaluations on diverse benchmarks are required.
> >
> > Unfortunately, my concerns mentioned in the weakness section have not been addressed in the author response. Thus, I would like to retain my negative score.

---

### Official Review · Reviewer_vZ4M · 2024-11-04

**Soundness:** 1
**Presentation:** 3
**Contribution:** 2
**Rating:** 3
**Confidence:** 4

**Summary:**

The paper presents an imitation learning / offline RL method that combines recent insights into the stable training of neural networks with cross-entropy losses and KL divergence minimization. The resulting agent predicts action sequences and achieves strong performance on the common D4RL benchmarking dataset.

**Strengths:**

The paper is mostly cleanly written and presents its insights well (with one notable exception below).

The experimental section is thorough, although I would encourage the authors to add some ablations.

**Weaknesses:**

Overall, the presentation of that paper leaves me unsure how the components of the loss are actually computed. The cross entropy formulation requires the actions to be binned, e.g. with the two-hot or HL-Gauss methods presented in Farebrother et al. However, in those formulations, the output of the action network would be a categorical. In this paper, the network outputs a Gaussian, which would not allow one to compute a cross entropy loss.

Another interpretation is that the authors don't use a categorical transformation, in which case their objective is not a cross entropy objective. It is unclear whether $y(a_t)$ is the probability of action $a_t$ (which would be necessary for the cross entropy objective) or simply the normalized action itself? In this case, how do the authors deal with the fact that $a_t$ is multidimensional? Their objective is then also mostly an MLE objective weighted by the size of the action, which has no clear probabilistic interpretation.The introduction of cross entropy which does not mention probabilities at all makes me think this might be the case?

From a theoretical angle, combining both a cross entropy loss and a KL minimization is odd, as both losses are mathematically equivalent, except for an additive constant which solely depends on the target. This means that adding a KL loss is simply equivalent to multiplying the cross entropy loss by 2. Writing out the two loss components together, this becomes somewhat obvious. $log (y(a_t))$ does not depend on the learned function, and the second part is equivalent to the (unweighted) cross entropy loss. When rearranging both losses, the added KL is equivalent to adding an offset of $\lambda$ to the return weighing factor.

While the paper talks about MLE, KL minimization (and therefore also cross-entropy) is closely related to MLE [1]. This makes the juxtaposition of the two somewhat confusing here. I assume the authors want to talk about the well-documented phenomenon that squared errors, corresponding to MLE under the assumption of a constant variance Gaussian, is an unsuitable loss for many neural network training setups. This should be clarified.

Reward weighted imitation learning has been proposed in the literature before, this should be acknowledged and differences discussed [2].

Similarly, predicting action sequences is well established in imitation learning, compare for example [3,4,5]. Again, this should be discussed.

3 seeds are well known to not result in statistically reliable results. Please follow established common practice here and report on a sufficiently large number of seeds. In addition, please report uncertainty estimates such as confidence or tolerance intervals. (compare [6] for advice)

All results seem to be reported on expert datasets. Given that the method is presented as an offline RL and not an IL method, it would be important to highlight if the learned policies can outperform the expert. I am unsure that the method would be able to exhibit relevant phenomena for strong offline RL such as trajectory stitching, given it's close reliance on IL objectives.

References:
[1] https://jaketae.github.io/study/kl-mle/
[2] Using reward-weighted imitation for robot reinforcement learning, J Peters, J Kober, 2009 IEEE Symposium on Adaptive Dynamic Programming and Reinforcement Learning
[3] Learning Fine-Grained Bimanual Manipulation with Low-Cost Hardware, Tony Z. Zhao, Vikash Kumar, Sergey Levine, Chelsea Finn, CORL 2023
[4] Diffusion Policy: Visuomotor Policy Learning via Action Diffusion, Cheng Chi, Zhenjia Xu, Siyuan Feng, Eric Cousineau, Yilun Du, Benjamin Burchfiel, Russ Tedrake, Shuran Song, IJRR 2023
[5]  RoboAgent - Towards Sample Efficient Robot Manipulation with Semantic Augmentations and Action Chunking,  Homanga Bharadhwaj, Jay Vakil, Mohit Sharma, Abhinav Gupta, Shubham Tulsiani, Vikash Kumar, ICRAA 2024
[6] https://arxiv.org/pdf/2304.01315

**Questions:**

n/a

---

> ### Author Response · Authors · 2024-11-23
>
> 1.Thank you for your comments and concerns regarding the cross-entropy loss computation and the use of Gaussian outputs in our model. We understand that the explanation of how our loss components are computed may have been unclear, and we appreciate the opportunity to clarify this.Thank you for your comments and concerns regarding the cross-entropy loss computation and the use of Gaussian outputs in our model. We understand that the explanation of how our loss components are computed may have been unclear, and we appreciate the opportunity to clarify this.To address your first point, while our neural network outputs both the mean and standard deviation of a Gaussian distribution, only the mean is used to compute the cross-entropy (CE) loss. The mean represents the central tendency of the predicted action distribution, which is treated as the action in our loss computation. The standard deviation, together with the mean, is used when calculating the Kullback-Leibler (KL) divergence to regularize the policy. We have now revised the manuscript to clarify this distinction and removed the more complex mathematical symbols and formulations to make the presentation clearer. The detailed, intuitive explanation and the specific implementation code are now included in the appendix, which we believe will further aid in understanding our approach.Regarding the issue of multidimensional actions, the neural network outputs a sequence of action vectors, where each action vector has dimensions of 2×action_dim×k, with k representing the number of discrete action bins. After generating epi_len/k action sequences, we concatenate these sequences to form a full action trajectory. Once the trajectory is formed, we perform normalization to ensure that the actions are appropriately scaled for the training process. We have added additional details and code examples in the appendix to address this specific concern.
>
> 2.As you pointed out, both cross-entropy and KL minimization are closely related, and their mathematical equivalence in some contexts can lead to confusion. We would like to clarify that cross-entropy loss and KL divergence are used in different stages of our training process and are not intended to be combined into a single objective.
>
> Cross-Entropy Loss: The CE loss is used during the training process to directly match the predicted action (mean of the Gaussian) to the target action, providing a stable gradient for the network. This loss is computed using only the mean output of the Gaussian distribution.
> KL Divergence: The KL divergence is used as a regularization term to ensure that the learned policy does not deviate too much from a prior distribution, and both the mean and standard deviation are involved in its calculation.
> Thus, these two losses serve different purposes: CE loss helps align the predicted actions with the target distribution, while KL divergence regularizes the network to maintain a reasonable exploration-exploitation balance. We have revised the paper to clearly differentiate these two objectives and explain their distinct roles.

---

> > ### Comment · Reviewer_vZ4M · 2024-11-23
> > **Reviewer reply**
> >
> > I took a look at the code and that raised significantly more questions. What is the interpretation of a softmax over Gaussian means? I genuinely do not think that this has any known mathematical validity (code line 26). This connects to my concern that fundamental statistical and mathematical quantities are treated in a highly non-standard way in this paper.

---

### Official Review · Reviewer_LiYP · 2024-11-05

**Soundness:** 1
**Presentation:** 2
**Contribution:** 2
**Rating:** 3
**Confidence:** 3

**Summary:**

This paper proposes Action Sequence Planner (ASP), a novel framework for offline reinforcement learning. ASP replaces the commonly used likelihood maximization loss in policy gradient methods with cross-entropy loss, resulting in stable training and consistent gradients. Rather than predicting actions step-by-step at each state, ASP directly predicts an entire action sequence using fully connected neural networks, avoiding the need for more complex architectures.

**Strengths:**

- The main idea of the paper—using cross-entropy loss to replace the commonly used likelihood loss—is both interesting and novel, with significant potential to open a new direction for offline RL.
- The novelty of the proposed approach is evident.
- The experimental results demonstrate the advantages of the proposed method effectively.

**Weaknesses:**

## Technical Issues
- The proposed method claims to address or relax the issue of inaccurate value estimation caused by using value functions to bootstrap future returns. However, throughout the paper, the only approach taken to achieve this seems to be the replacement of the value/advantage function with the discounted return, which appears insufficient. As discussed in work like GAE [1], multiple design choices could serve as learning objectives or weigh the action reproducing likelihood. The use of a value function, rather than the true return, helps avoid high variance caused by different rollouts. Directly using the true return may increase variance, potentially harming training stability, which weakens the technical justification.

- From a deep learning perspective, using fully connected (FC) layers to directly generate trajectories or sequences is not technically sound, due to their poor scalability to sequence length, with complexity $O(N^2)$. Generally, sequence models like RNNs and Transformers are preferred for sequence modeling. The proposed method contradicts this intuition without providing strong arguments or experiments to justify why FC layers are chosen over alternative models that might better fit sequence modeling tasks.

- The mapping from the action trajectory to its distribution in line 189 may lead to constant shift risks. For instance, if a constant is added to all actions in a trajectory, the distribution remains unchanged. Using such a distribution in the learning objective could introduce a constant shift in the learned action trajectory. Please analyze this potential risk.

- The trajectory distribution calculation in line 189 does not account for the dimensionality of actions. For tasks with high-dimensional action spaces, how should this distribution be computed?

- The content between lines 227 and 230 is unclear. Please consider restructuring it for clarity.

- Line 231, if $y_t(a)$ is a scaling factor that adjusts each action’s contribution to the gradient update, does this mean that the action with the minimum value will always have zero contribution, as its corresponding $y(t)$ would be zero?

- In line 236, why does incorporating actions that are deemed more significant by the normalized distribution benefit the policy? The causality here is unclear, as the distribution is merely a numerical representation of the action value, not a meaningful distribution that captures the reproducing capability. Please provide further explanation.

- In line 251, why is an off-policy method mentioned here?

- The proposed method, particularly the trajectory distribution matching component, is technically quite similar to Behavior Cloning (although BC is imitation learning rather than offline RL). What causes the ablated method, ASP-MLE, to consistently perform worse than BC in Table 1? Is the setting for ASP-MLE fair, given that it uses hyperparameters from a model trained with a different loss function? Notably, the gradient magnitude differs by $10^4$?

## Writting
The storyline of the paper feels somewhat loose and could benefit from proofreading. Many arguments and conclusions are presented without sufficient supporting evidence or references to relevant literature.

## Missing discussions to literature works
Several RL papers [2, 3] relate to action sequence generation and might offer alternative design choices beyond using FC layers. Please consider discussing and comparing these methods.

## Minor
Please number key equations in the paper for readability.


## References
[1] Schulman, John, et al. "High-dimensional continuous control using generalized advantage estimation." ICLR 2016.

[2] Raffin, Antonin, Jens Kober, and Freek Stulp. "Smooth exploration for robotic reinforcement learning." Conference on robot learning. PMLR, 2022.

[3] Zhang, Haichao, Wei Xu, and Haonan Yu. "Generative planning for temporally coordinated exploration in reinforcement learning." ICLR 2022.

**Questions:**

Please address the issues listed in the weaknesses section.

---

> ### Author Response · Authors · 2024-11-23
>
> 1.Thank you for your insightful comment regarding the use of discounted return as a solution for inaccurate value  estimation.  We acknowledge the concern that directly using discounted returns,  without a more sophisticated value function,  may result in higher variance and thus affect training stability.  However, our approach,  which replaces the value/advantage function with discounted returns, serves as an initial,  simplified step toward addressing the challenges posed by inaccurate value estimation and error accumulation.
> 2.We appreciate your concern regarding the use of fully connected (FC) layers to directly generate action sequences. While it is true that sequence models like RNNs and Transformers are generally preferred for sequence modeling due to their ability to capture long-range dependencies, our choice of FC layers was motivated by the goal of providing a simple and computationally efficient baseline for our method. In our experiments, this approach yielded stable results with relatively low time cost during training, which is particularly important for verifying the effectiveness of our proposed loss function. We also recognize that sequence models such as Transformers and RNNs might perform better in capturing temporal dependencies in more complex scenarios. We plan to extend our method in future work by incorporating sequence models like RNNs or Transformers, and will validate these alternatives in subsequent experiments. For now, our focus has been on establishing a straightforward approach that provides a solid foundation for further exploration.
> 3.We agree that if a constant is added to all actions in a trajectory, it could lead to an unchanged distribution, which may pose issues during training. To mitigate this, we apply a normalization step to the action distribution, ensuring that the trajectory distribution is adjusted appropriately, and that the learned action sequence remains meaningful.
> 4.We acknowledge your concerns about the clarity of our notation and presentation. To improve the readability and comprehension of our method, we have removed the mathematical formulas and notations in Section 3.2. Instead, we have included a more detailed explanation of the method in the appendix, along with its implementation code. We believe this change provides a more intuitive and accessible way for readers to understand our approach.
> 5.Regarding the performance of ASP-MLE compared to BC, the lower performance of ASP-MLE in our experiments is indeed due to the instability introduced by using the maximum likelihood estimation loss on entire action sequences. As illustrated in the gradient magnitude plots, this instability results in erratic gradients and can lead to incorrect updates during training. In contrast, our method with cross-entropy loss provides smoother gradient updates, leading to more stable training and better performance.

---

> > ### Comment · Reviewer_LiYP · 2024-11-25
> > **Several issues are not addressed or discussed**
> >
> > Hi,
> >
> >
> > Thank you for your reply and updates to the paper.
> >
> > Unfortunately, I found that most of my concerns were not addressed well or were outright ignored, including:
> >
> > **Discounted Return vs. Value Function**:
> >
> > The reasoning behind using discounted return instead of the value function remains unclear. The value function is a fundamental component of RL models, yet your response merely repeated your design choices without providing any convincing justification.
> >
> > **Sequence Models vs. FC Layers:**
> >
> > While using fully connected (FC) layers for prototyping is acceptable, failing to incorporate or compare sequence models (e.g., RNNs, Transformers) is a major shortcoming for a methodology that focuses on a sequence of actions. This is especially critical given the presence of prior offline RL work that integrates sequence models effectively.
> >
> > **High-Dimensional Action Space:**
> >
> > My concerns regarding the handling of high-dimensional action spaces were not addressed or even discussed.
> >
> > **Mathematical Content in Section 3.2:**
> >
> > You stated that the outdated mathematical content and discussion in Section 3.2 was moved to the appendix, but I could not find it in the updated paper's appendix.
> >
> > **Content Deletions Without Notice:**
> >
> > - Significant content that raised concerns has been quietly removed from the paper without any acknowledgment, such as the scaling factor $y_t(a)$.
> >
> >
> > **Line 236–238 Issues:**
> > - My concerns regarding the content in lines 236 to 238 remain completely unaddressed and unexplained.
> >
> > Given these unresolved issues and the lack of engagement with my feedback, I maintain my negative opinion for this paper.

---

### Meta-Review · Area_Chair_RHKX · 2024-12-21

**Metareview:**

This paper introduces an offline RL method that utilizes cross entropy loss in policy gradient methods together with action-sequence planning. Experimental results suggest superior performance on D4RL benchmarking datasets.

The paper seems incomplete in many different ways. Technically, the work appears problematic as it incorrectly contrasts cross-entropy loss with maximum likelihood loss, despite the fact that the former can be derived as a specific case of the latter. At a higher level, the connection between the motivation and the approach of this paper is not clear. The abstract starts with the problem of instability in offline RL but then proposes action-sequence planning as a solution to this issue without much reasoning about how they are connected. I strongly recommend revising the paper from a third-person and first-principles perspective to understand what relevant problem the proposed method is addressing and motivating the work plainly from that perspective.

**Additional Comments On Reviewer Discussion:**

The reviewers pointed out the above and several other issues with the paper, including technical details. The empirical results are limited to D4RL and a custom budget allocation task; broader benchmarks and rigorous ablations are missing. Overall, the paper simply does not seem to be ready for this venue. We look forward to a substantially improved version of this paper in a future venue.

---

### Decision · Program_Chairs · 2025-01-22

Reject